# Adolescent sexual and reproductive health literacy in Saudi Arabia: The roles of school type and maternal education

Asma Alshanqiti[1]*, Reem F. Al-Mughthawi[2], Elaf Aljabri[2], Aisha R. Al-Rashidi[2], Fatema A. Saleh[2], Raghdah Alrehaili[2], Taif N. Alahmadi[2], Mohamed wagdy[3]

**1** Department of Family and Community Medicine and medical education, College of Medicine, Taibah University, Medina, Saudi Arabia, **2** College of Medicine, Taibah University, Medina, Saudi Arabia, **3** Faculty of Medicine, Modern university for Technology and Information, Cairo, Egypt

* amshanqiti@taibahu.edu.sa

## Abstract

### Background

Adolescent sexual and reproductive health (SRH) literacy remains uneven in conservative settings. We assessed SRH knowledge among female secondary-school students in Al-Madinah Al-Monawara and examined socio-demographic correlates and parent–daughter communication.

### Methods

Cross-sectional survey of 389 students (grades 10–12) selected via stratified cluster sampling from government and private schools (20 June–20 July 2024). A validated Arabic questionnaire measured domain-specific SRH knowledge (menstruation, reproductive anatomy, contraception, sexual relationships), sexually transmitted disease (STD) awareness, and parent–daughter communication. Analyses included $t$-tests/ANOVA, Pearson correlations, and multivariable OLS with HC3-robust SEs.

### Results

Knowledge gaps were concentrated in contraception and sexual relationships; 61 students incorrectly endorsed casual contact as a transmission route for STDs. Private-school students had higher composite SRH scores than government-school peers ($t$-test, $p < 0.01$). Maternal education correlated with SRH knowledge ($r = 0.38$, $p < 0.001$). Scores varied by age (ANOVA $F = 4.67$, $p = 0.009$), but age was not an independent predictor in regression. In multivariable models, private-school attendance ($\beta = 0.212$, $p = 0.018$) and higher maternal education ($\beta = 0.043$, $p = 0.019$) independently predicted higher SRH knowledge; paternal education, age, and family size were non-significant. Parent–daughter discussions were infrequent; 72% of students supported school-based SRH education.

**Data availability statement:** All relevant data underlying the results of this study are available within the paper and its Supporting Information files. The anonymized dataset used for the analyses and the full Arabic questionnaire employed in the survey are provided as supplementary materials to ensure transparency and reproducibility of the study.

**Funding:** The author(s) received no specific funding for this work.

**Competing interests:** The authors have declared that no competing interests exist.

## Conclusions

SRH literacy among Saudi adolescent girls is uneven, with prominent misconceptions in contraception and transmission. School type and maternal education are actionable levers. Culturally attuned, standardized curricula—especially in government schools—combined with parent-focused communication supports are warranted to close knowledge gaps and reduce stigma.

## 1. Introduction

Adolescence is a transitional life stage marked by rapid physical, psychological, and social changes, during which sexual and reproductive health (SRH) knowledge plays a pivotal role in shaping lifelong outcomes [1,2]. Without access to accurate and age-appropriate information, adolescents face higher risks of unintended pregnancy, sexually transmitted infections (STIs), stigma, and anxiety about normal physiological changes. SRH literacy, defined as the ability to access, understand, appraise, and apply health information in reproductive contexts, is recognized globally as a cornerstone of adolescent wellbeing [3,4].

In conservative societies, including Saudi Arabia, SRH remains a culturally sensitive subject. Taboos and discomfort limit open discussion within schools and families, resulting in adolescents turning to peers or online sources that may be inaccurate. Qualitative research among Saudi women highlights that many reported entering puberty with little guidance, often experiencing confusion and negative emotions [5,6]. Parents and teachers avoided conversations or shared partial or misleading information, leaving young women with significant uncertainty about menstruation and reproductive processes [2].

Quantitative evidence confirms these knowledge gaps. A study of 419 female students in Riyadh reported that among those aged ≤ 15 years, 54.1% had poor SRH knowledge, and this proportion worsened to 70.7% among those older than 15 years. Parental education and inclusion of sexual health content in school curricula were key predictors of better knowledge, suggesting structural and familial influences are both critical [2]. Intervention research provides encouraging evidence that knowledge can be improved: in Qassim, after a structured educational program, recognition of intrauterine devices as contraceptives rose from 27.2% before intervention to 67.6% afterwards. These findings demonstrate both the scale of misconceptions and the potential for educational initiatives to achieve rapid improvement when carefully delivered [7,8].

Parental engagement, particularly parent–adolescent communication, strongly influences SRH outcomes. A meta-analysis of 71 studies concluded that more frequent parent–child discussions are significantly associated with safer sexual behaviors, including increased contraceptive use and delayed initiation of sexual activity [8]. Yet, the frequency and comfort of these discussions are often low, particularly in conservative or resource-limited contexts. In Ethiopia, only 30.6% of adolescents reported discussing at least 60% of core SRH topics with their

parents [4]. In Nepal, 57% of secondary students reported no SRH communication with parents at all, and fewer than 20% discussed more than one issue [9]. These trends mirror findings across the Middle East, where parents express concern that conversations might encourage early sexual experimentation, thereby limiting both timing and content [6,10,11].

Maternal education consistently emerges as a particularly strong predictor of adolescent SRH literacy. Mothers with higher education levels are more likely to provide accurate information and to initiate conversations, while adolescents of less educated mothers frequently rely on peers or unreliable sources. International research shows that where schools and parents both provide accurate input, adolescents achieve higher literacy, suggesting synergistic effects between school and family [5]. In Saudi Arabia, differences between public and private schools further complicate the picture: private schools often have greater flexibility in health education content and resources, while public schools may offer limited or inconsistent instruction [2,12,13].

Taken together, current evidence indicates three recurring patterns. First, SRH knowledge among adolescents is uneven and tends to be particularly weak in domains such as contraception and STI prevention. Second, structural factors such as school type and the presence of formal curriculum content significantly influence knowledge acquisition. Third, family factors, especially maternal education and communication practices, act as additional determinants that can either reinforce or undermine school learning [14].

Despite this, substantial gaps remain in the Saudi literature, particularly for regions such as Al-Madinah. Few studies jointly examine school type, maternal and paternal education, and parent–daughter communication in relation to domain-specific knowledge. There is also limited quantitative evidence on the comfort level of these conversations, perceived barriers, and the extent to which cultural taboos constrain them. Addressing these gaps is important, as adolescence represents a window where timely interventions can pre-empt misinformation and its long-term consequences [15,16].

The present study, therefore, assesses domain-specific SRH knowledge among female secondary school students in Al-Madinah Al-Monawara, while evaluating socio-demographic correlates with emphasis on school type and parental education. It also investigates patterns of parent–daughter communication, including frequency, comfort, and barriers, as potential levers for intervention. Building on prior evidence, it is expected that knowledge gaps will be most pronounced in contraception and STI prevention, that private school attendance and higher maternal education will predict higher literacy, and that frequent, comfortable parental communication will correlate positively with scores. In Saudi Arabia, adolescence spans 10–19 (WHO). This study, however, targeted students aged ≥18 due to ethical and administrative considerations in researching sensitive topics among minors and the relevance to imminent reproductive decision-making. The sample was restricted to female students, reflecting gender-segregated education and the culturally predominant maternal role in conveying reproductive health information to daughters. This focus facilitated context-specific examination of mother–daughter communication, while acknowledging limited generalizability due to the exclusion of male adolescents. By identifying these associations, the study aims to inform culturally sensitive strategies that can strengthen adolescent SRH literacy in Saudi Arabia.

## 2. Materials and methods

### 2.1. Study design and setting

This study employed a cross-sectional design to evaluate sexual and reproductive health (SRH) knowledge, including awareness of STDs, transmission routes, prevention methods, and the role of parental education and communication in shaping adolescents' understanding. The study was conducted in Al-Madinah Al-Monawara, Saudi Arabia, targeting female secondary school students from both government and private institutions. Data collection was carried out between 20 June 2024 and 20 July 2024, ensuring a representative sample of students across varied socio-economic and educational backgrounds.

 

## 2.2. Study population and sample size

The study population comprised female students aged 18 years and above enrolled in government and private secondary schools in Al-Madinah Al-Monawara. This age group was selected due to its relevance to reproductive health decision making, especially given its legal and social significance in the Saudi Arabian context.

A total of 389 students participated in the study, meeting the final sample size requirement. The sample was determined using a 95% confidence level ($\alpha = 0.05$) and a 5% margin of error, calculated through standard sample size determination formulas. A stratified random sampling technique was employed to ensure balanced representation across different school types (government vs. private) and socio-economic backgrounds. Schools were randomly selected, and students were recruited voluntarily, maintaining a diverse and representative sample of the adolescent population.

## 2.3. Study instrument and data collection

A self-administered electronic questionnaire was developed in Arabic, using validated tools from previous studies to assess SRH knowledge, misconceptions, and communication patterns [1,2]. The questionnaire was pre-tested on 30 students to evaluate clarity, reliability, and validity, and minor modifications were made based on feedback before full-scale distribution.

The questionnaire was structured into four key sections:

1. Socio-Demographic Characteristics

o Age, nationality, school type (government/private), family size, and parental education levels.

2. Assessment of SRH Knowledge

o Awareness of menstruation, pregnancy, contraception, and sexually transmitted diseases (STDs).
 o Identification of common misconceptions related to reproductive health.
 o Sources of SRH knowledge and the perceived reliability of these sources.

3. STD Awareness, Transmission, and Prevention Knowledge

o Knowledge of HIV/AIDS, syphilis, gonorrhea, and chlamydia.
 o Understanding of correct and incorrect modes of STD transmission, including misconceptions regarding casual contact, blood transmission, and sexual activity.
 o Awareness of STD prevention strategies, such as condom use, monogamy, and abstinence.

4. Mother–Daughter Communication on SRH

o Frequency and quality of discussions between adolescents and their mothers.
 o Topics covered, including menstruation, pregnancy, contraception, STDs, and sexual harassment.
 o Initiator of the conversation and the communication style (interactive vs. one-sided).
 The questionnaire was distributed electronically via an online survey platform, ensuring anonymity and privacy to encourage honest responses and reduce social desirability bias.

## 2.4. Data management and statistical analysis

All 389 responses were cleaned, coded, and analyzed using IBM SPSS (latest version). A combination of descriptive and inferential statistical methods was used to examine SRH knowledge, socio-demographic disparities, and communication barriers:

1. Descriptive Statistics

o Frequencies and percentages were calculated for categorical variables.
   o Means and standard deviations (SD) were determined for continuous variables.

2. Bivariate Analysis

o Chi-square ($\chi^2$) test was used to examine associations between SRH knowledge and socio-demographic factors (e.g., school type, parental education, and mother–daughter communication frequency).
   o Independent *t*-tests were applied to compare SRH knowledge scores between government and private school students.
   o One-way ANOVA tests were conducted to examine variations in SRH knowledge across different age groups and school types.

3. Multivariable linear regression (Ordinary Least Squares, OLS) was conducted to identify independent predictors of composite SRH knowledge scores. HC3 robust standard errors were applied to address potential heteroskedasticity. The composite knowledge score was treated as a continuous outcome variable. Covariates included age, school type, maternal education, paternal education, and family size.

4. Correlation Analysis

   o Pearson correlation coefficients (*R*-values) were computed to assess the relationships between maternal education, SRH knowledge scores, and parent–adolescent communication patterns.

5. Statistical Significance

o A $p < 0.05$ was considered statistically significant for all analyses.

## 2.5. Ethical considerations

Ethical approval for this study was obtained from the Institutional Review Board of Taibah University and registered at the US Department of Health and Human Services, IORG0008716 – IRB00010413—approval date: 20 March 2024.

• Informed written consent was obtained from all 389 participants before completing the questionnaire.

• Parental written consent was required for students under 18 years old.

• Confidentiality and anonymity were strictly maintained, with all responses de-identified

   before analysis.
   • Participants were informed that their participation was voluntary, and they had the right
   to withdraw from the study at any time without penalty.

• All collected data was securely stored, with restricted access limited to authorized

researchers to maintain participant privacy. The anonymized dataset and full study questionnaire are provided as supple-mentary materials accompanying this manuscript.

## 3. Results

The analysis identified statistically significant socio-demographic and socio-economic disparities in SRH knowledge, influenced by age, nationality, school type, parental education, and mother–daughter communication. A one-way ANOVA test (*F = 4.67, p = 0.009*) confirmed a significant difference in SRH knowledge scores across age groups, with older participants

(*> 19 years*) exhibiting higher scores than younger ones (Fig 1). A t-test (*p = 0.003*) indicated a significant disparity between students from private and government schools, reinforcing the role of school type in determining SRH literacy.

Parental education was another key determinant, with a Pearson correlation test (*r = 0.38, p < 0.01*) showing a moderate positive relationship between maternal education and SRH knowledge scores. As seen in Table 1, participants with highly educated mothers (Master's degree) had the highest SRH knowledge scores (62.40 ± 4.90), compared to those whose mothers had only primary education (61.10 ± 5.40). Furthermore, frequent mother–daughter communication was associated with significantly higher knowledge scores (*p < 0.01*), highlighting the critical role of family discussions in reproductive health awareness.

The results indicate that students from government schools, lower-income families, and households with limited parental education require targeted interventions to bridge knowledge gaps. Strengthening structured SRH curricula in public schools, promoting parental engagement programs, and expanding school-based interventions are essential to ensure equitable access to SRH education among adolescents.

Table 1 presents the socio-demographic factors influencing SRH knowledge scores, including age, nationality, school type, family size, and parental education levels. The table also highlights subgroup disparities, emphasizing the role of school type, parental education, and mother–daughter communication in shaping SRH knowledge.

As shown in (Fig 1). The distribution of participants across age groups and their corresponding SRH knowledge scores. The 18–19 age group had the highest representation, while the > 19 age group demonstrated the highest knowledge scores.

Age and School Type: Older participants and those from private schools exhibited significantly better SRH knowledge, emphasizing the need for age-appropriate SRH interventions and improvements in public school curricula.

Parental Education: Higher parental education and frequent communication were associated with better knowledge scores, highlighting the importance of family engagement in reproductive health discussions.

Targeted Interventions: The observed disparities indicate the need for policies that strengthen SRH education in government schools and encourage parental involvement, ensuring that all adolescents have equitable access to reproductive health literacy.

Parental education varied, with 30.85% of fathers and 28.28% of mothers having secondary education. Higher parental education levels were associated with slightly better SRH knowledge scores, as seen in the mean SRH knowledge scores

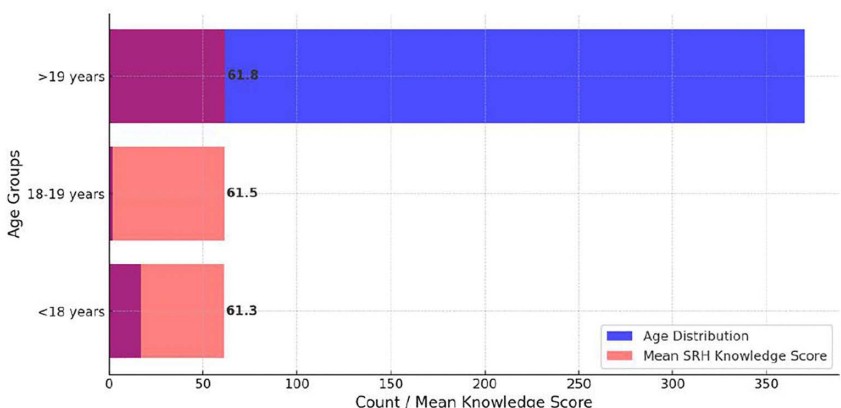

**Fig 1. Distribution of Age Groups and SRH Knowledge Scores.** This figure illustrates the distribution of participants across age groups and their corresponding sexual and reproductive health (SRH) knowledge scores, showing higher scores among older students.

**Table 1. Socio-demographic characteristics & socio-economic disparities in SRH knowledge.**

| Variable | Categories | Fre-quency (n) | Percentage (%) | Mean SRH Knowl-edge Score (SD) | Subgroup Factor | Mean SRH Knowledge Score (Lower Group) | Mean SRH Knowledge Score (Higher Group) | p-value |
|---|---|---|---|---|---|---|---|---|
| Age Group | <18 years/ 18–19 years/>19 years | 88/ 188/ 113 | 22.6%/ 48.3%/ 29.0% | 61.29 (±5.53)/ 61.50 (±0.71)/ 61.81 (±5.33) | | | | |
| Nationality | Saudi/ Non-Saudi | 250/ 139 | 64.27%/ 35.73% | 61.75 (±5.10)/ 62.15 (±5.25) | | | | |
| School Type | Government/ Private | 220/ 169 | 56.55%/ 43.45% | 61.50 (±4.95)/ 62.30 (±5.40) | School Type (Gov-ernment vs. Private) | 61.50 (±4.95) | 62.30 (±5.40) | <0.01 |
| Family Size | 3–4 persons/ 5 or more persons | 160/ 229 | 41.13%/ 58.87% | 62.00 (±5.12)/ 61.85 (±5.20) | | | | |
| Father's Education Level | Illiterate/ Primary/ Secondary/ Higher Secondary/ Mas-ter's Degree | 30/ 70/ 120/ 100/ 69 | 7.72%/ 17.99%/ 30.85%/ 25.71%/ 17.73% | 60.50 (±5.65)/ 61.20 (±5.45)/ 61.75 (±5.30)/ 62.10 (±5.15)/ 62.50 (±4.98) | Father's Education (Higher vs. Lower) | 60.50 (±5.65) | 62.50 (±4.98) | <0.05 |
| Mother's Education Level | Illiterate/ Primary/ Secondary/ Higher Secondary/ Mas-ter's Degree | 40/ 75/ 110/ 90/ 74 | 10.28%/ 19.28%/ 28.28%/ 23.14%/ 18.97% | 60.80 (±5.50)/ 61.10 (±5.40)/ 61.65 (±5.25)/ 62.00 (±5.10)/ 62.40 (±4.90) | Mother's Education (Higher vs. Lower) | 61.10 (±5.40) | 62.40 (±4.90) | <0.05 |
| Mother-Daughter Communi-cation | Frequent vs. Infrequent | – | – | – | Communi-cation (Fre-quent vs. Infrequent) | 61.10 (±5.30) | 62.40 (±5.20) | <0.01 |
| Mean SRH Knowledge Score (Overall) | – | – | – | **61.98 (±5.18)** | | | | |

for participants whose parents had a Master's degree (fathers: 62.50±4.98, mothers: 62.40±4.90). The overall mean SRH knowledge score was 61.98±5.18, with minor differences across age groups, school type, and parental education levels. Fig 1 illustrates the distribution of age groups and corresponding SRH knowledge scores, highlighting the age-related variations in knowledge levels. The findings suggest that while socio-demographic factors influence SRH knowledge, further investigation is needed to assess their long-term impact on reproductive health awareness (Fig 2).

### 3.2. Sources of SRH Knowledge & Parent–Child Communication

The primary sources of SRH knowledge were school type and parental education levels, both of which significantly influenced SRH knowledge scores Table 2. Private school students (43.45%) had slightly higher SRH knowledge scores (62.30±5.40) compared to those in government schools (56.55%), with scores of 61.50±4.95.

Parental education also played a crucial role, with participants whose parents had a Master's degree demonstrating the highest SRH knowledge scores (Father: 62.50±4.98, Mother: 62.40±4.90). Conversely, participants with illiterate parents had lower SRH knowledge scores (Father: 60.50±5.65, Mother: 60.80±5.50), emphasizing the importance of parental education in reproductive health awareness.

Regarding parent–child communication, discussions on SRH topics were infrequent:

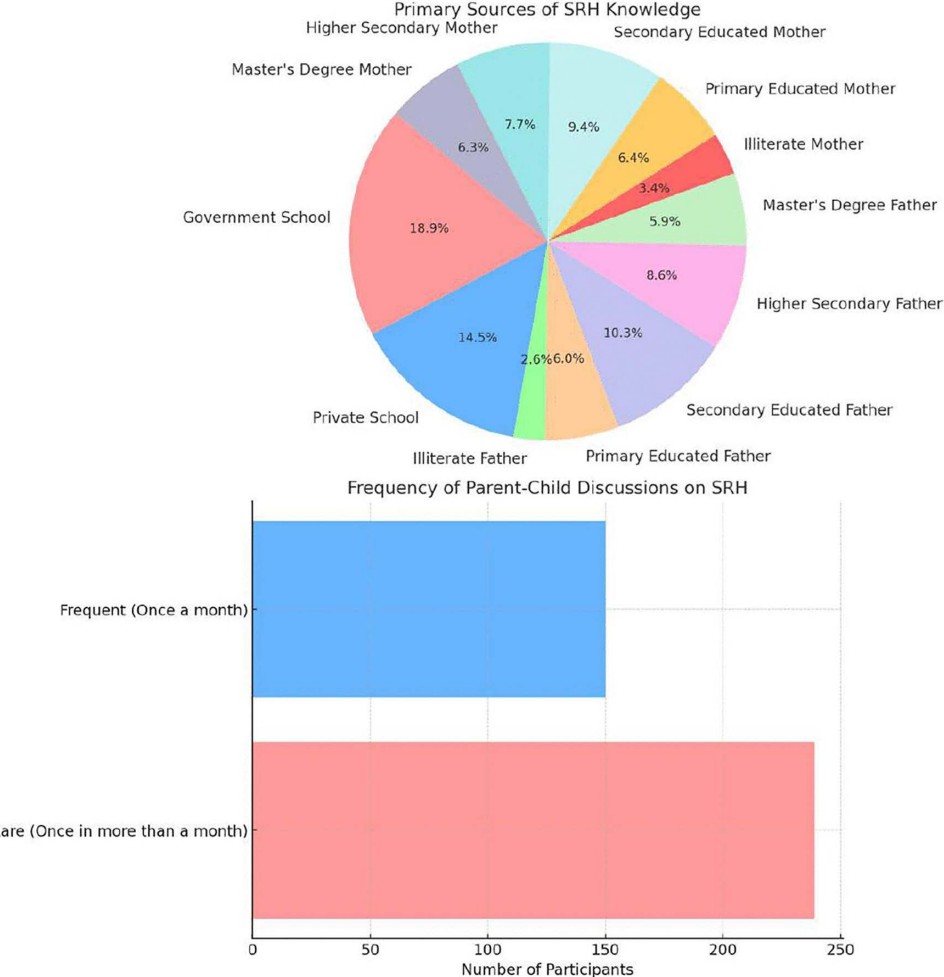

**Fig 2. Presents the distribution of SRH knowledge sources (Pie Chart) and the frequency of parent-child discussions (Bar Chart), highlighting the limited role of parental discussions in SRH education.** Knowledge and Misconceptions about SRH.

- 38.5% of participants reported frequent discussions (*at least once per month*).

- 61.5% reported rare discussions (*less than once per month*).

### 3.3. Knowledge and misconceptions about SRH

This study assessed knowledge and misconceptions regarding menstruation, reproductive systems, contraception, and sexual relationships among female students. The findings revealed significant knowledge gaps, particularly in contraception and sexual relationships, where over 60% of students had misconceptions. A one-way ANOVA test ($F = 231.60$, $p < 0.001$) confirmed a statistically significant difference in knowledge levels across these domains. The mean knowledge scores varied across topics, with menstruation knowledge scoring the highest ($2.39 \pm 0.67$) with only 10.8% misconceptions, while sexual relationship knowledge scored the lowest ($1.28 \pm 0.45$), with 72.2% misconceptions. The misconception rate for contraception (62.2%) highlights widespread misinformation regarding pregnancy prevention methods, while 45.0% of students lacked proper understanding of reproductive systems.

**Table 2. Sources of SRH Knowledge & Parent-Child Communication.**

| Source | Category | Frequency (n) | Mean SRH Knowledge Score (SD) |
|---|---|---|---|
| School Type | Government | 220 | 61.50 (±4.95) |
| | Private | 169 | 62.30 (±5.40) |
| Father's Education Level | Illiterate | 30 | 60.50 (±5.65) |
| | Primary | 70 | 61.20 (±5.45) |
| | Secondary | 120 | 61.75 (±5.30) |
| | Higher Secondary | 100 | 62.10 (±5.15) |
| | Master's Degree | 69 | 62.50 (±4.98) |
| Mother's Education Level | Illiterate | 40 | 60.80 (±5.50) |
| | Primary | 75 | 61.10 (±5.40) |
| | Secondary | 110 | 61.65 (±5.25) |
| | Higher Secondary | 90 | 62.00 (±5.10) |
| | Master's Degree | 74 | 62.40 (±4.90) |

Table 3 shows a regression analysis that identified the type of school and maternal education as significant predictors of SRH knowledge. Private school students had significantly higher SRH knowledge ($\beta = 0.212, p = 0.018$), suggesting better curriculum-based exposure. Additionally, higher maternal education was positively associated with SRH knowledge ($\beta = 0.043, p = 0.019$), reinforcing the role of parental influence in knowledge acquisition. However, age, family size, and father's education were not statistically significant predictors, indicating that schooling and maternal education play more vital roles in shaping SRH awareness.

To visualize the distribution of misconceptions, Fig 3. Shows a line chart illustrating the percentage of students with incorrect responses in each SRH category. Sexual relationship knowledge exhibits the highest misconceptions (72.2%), followed by contraception (62.2%), while menstruation knowledge shows the least misconceptions (10.8%), reinforcing the need for targeted education in contraception and sexual health discussions.

Overall, the findings highlight the urgent need for enhanced SRH education, especially in contraception and sexual health awareness. The statistically significant impact of school type and maternal education suggests that structured education and parental engagement are key factors in improving adolescent SRH knowledge. Future interventions should

**Table 3. Knowledge Scores, Misconceptions, and Predictors of SRH Knowledge.**

| Category | Mean±SD | Min Score | Max Score | % Misconceptions (<2) | Regression Coefficient (β) | p-value | Significance |
|---|---|---|---|---|---|---|---|
| Menstruation Knowledge | 2.39±0.67 | 1.0 | 3.0 | 10.8% | – | – | – |
| Reproductive Systems Knowledge | 1.71±0.73 | 1.0 | 3.0 | 45.0% | – | – | – |
| Contraception Knowledge | 1.46±0.64 | 1.0 | 3.0 | 62.2% | – | – | – |
| Sexual Relationship Knowledge | 1.28±0.45 | 1.0 | 2.0 | 72.2% | – | – | – |
| Predictors of SRH Knowledge (Regression Analysis) | | | | | | | |
| Type of School (Private vs. Public) | – | – | – | – | 0.2117 | 0.018 | Significant |
| Mother's Education Level | – | – | – | – | 0.0427 | 0.019 | Significant |
| Age | – | – | – | – | 0.0196 | 0.464 | Not Significant |
| Family Size | – | – | – | – | −0.1097 | 0.122 | Not Significant |
| Father's Education Level | – | – | – | – | −0.0168 | 0.404 | Not Significant |

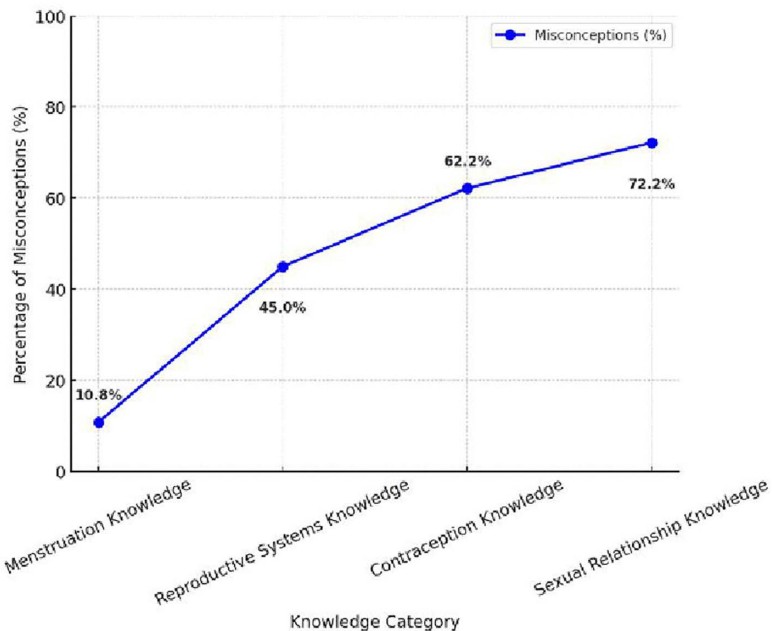

**Fig 3. Misconceptions in SRH Knowledge.** This figure displays the percentage of misconceptions across SRH domains, with the highest misconceptions observed in sexual relationships and contraception.

focus on comprehensive curriculum reforms and culturally appropriate awareness programs to bridge the knowledge gap and reduce misconceptions.

### 3.4. Comprehensive awareness of STDs, transmission methods, and prevention

This study examined students' awareness of STDs, transmission methods, and prevention strategies, revealing significant disparities in knowledge levels. A one-way ANOVA test (F = 16.11, p < 0.001) confirmed that awareness significantly varied across different domains, with transmission knowledge being the highest, while disease-specific awareness and prevention strategies lagged S4 Table.

The general awareness of STDs had a mean score of 2.04 ± 0.98, indicating moderate knowledge, but specific diseases such as chlamydia had extremely low recognition (only 19 students identified it correctly). The most well-known STD was HIV/AIDS (158 students identified it), followed by syphilis (136 students). However, a substantial proportion of students lacked knowledge about gonorrhea (76 students) and chlamydia, highlighting major educational gaps in bacterial STDs (Fig 4).

A deeper analysis of STD transmission knowledge showed that mother-to-child transmission was the most widely recognized mode of transmission (143 students), followed by blood contact (78 students). However, 61 students incorrectly believed that STDs could spread through casual contact, reinforcing the presence of misconceptions that could contribute to social stigma.

Regarding STD prevention strategies, abstinence and monogamy were the most frequently cited preventive measures (159 students), followed by condom use (119 students). However, a notable gap was observed in the understanding of regular health check-ups and partner screening, which are critical in preventing STDs. A comparative analysis of transmission knowledge and prevention awareness (Fig 5) reveals that while students had some understanding of transmission routes, their knowledge of prevention strategies remained inadequate (Table 4).

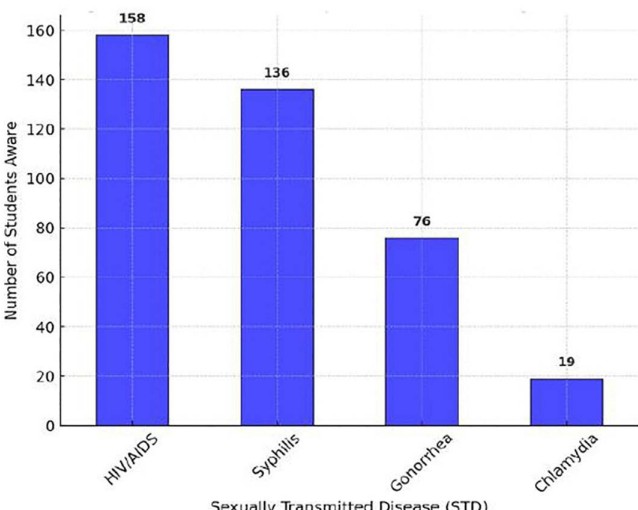

**Fig 4. Awareness of Specific STDs among Students.** This figure shows the level of awareness of different sexually transmitted diseases (STDs), with HIV/AIDS being the most recognized and chlamydia the least recognized.

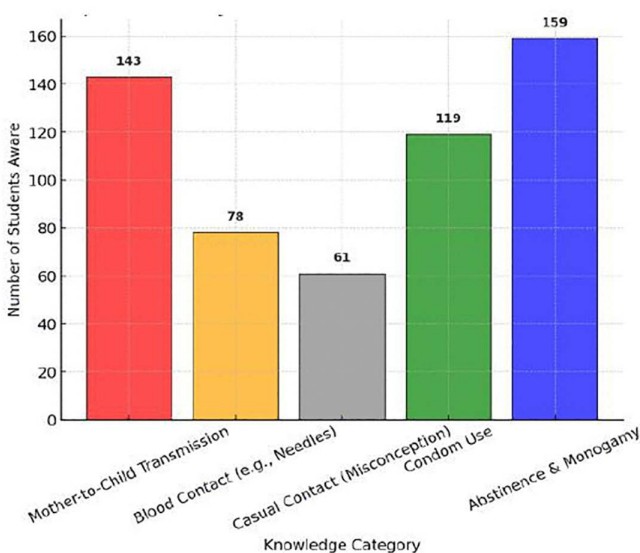

**Fig 5. Comparative Analysis of STD Transmission and Prevention Knowledge.** This figure compares students' knowledge of STD transmission routes and prevention strategies, demonstrating stronger understanding of transmission than prevention.

To further analyze misconceptions and knowledge disparities, Figs 4. and 5. provide comparative visualizations of awareness levels.

• Fig 4. shows that HIV/AIDS had the highest recognition (158 students), while bacterial STDs like *Chlamydia* (19 students) were the least recognized. This suggests that education efforts have primarily focused on viral STDs, leaving bacterial infections underrepresented.

**Table 4. Comprehensive STDs Awareness, Transmission, and Prevention Knowledge.**

| Category | Mean ± SD | Min Score | Max Score | ANOVA p-value | Correlation with Other Domains |
|---|---|---|---|---|---|
| STDs Awareness (General) | 2.04 ± 0.98 | 1.0 | 4.0 | < 0.0001 | 0.34 |
| HIV/AIDS Awareness | 158 identified | – | – | – | – |
| Syphilis Awareness | 136 identified | – | – | – | – |
| Gonorrhea Awareness | 76 identified | – | – | – | – |
| Chlamydia Awareness | 19 identified | – | – | – | – |
| STD Transmission Knowledge | 2.41 ± 1.05 | 1.0 | 4.0 | < 0.0001 | 0.27 |
| Mother-to-Child Transmission | 143 identified | – | – | – | – |
| Blood Contact (e.g., Needles) | 78 identified | – | – | – | – |
| Casual Contact (Misconception) | 61 identified | – | – | – | – |
| STD Prevention Awareness | 2.10 ± 0.84 | 1.0 | 3.0 | < 0.0001 | 0.27 |
| Condom Use | 119 identified | – | – | – | – |
| Abstinence & Monogamy | 159 identified | – | – | – | – |

- Fig 5. highlights knowledge gaps in transmission and prevention, where students had better knowledge of how STDs spread but lacked awareness of prevention strategies beyond abstinence and condoms.

HIV/AIDS awareness is relatively high, but bacterial STDs remain poorly understood. Students may recognize the concept of STDs but lack disease-specific knowledge, leading to gaps in symptom identification and prevention strategies. Misconceptions about casual transmission must be addressed, as they contribute to social stigma and misinformation. Education programs should extend beyond abstinence and condom use to include a broader understanding of sexual health interventions, including regular testing, partner screening, and HPV vaccination. Future interventions should close the gap between transmission awareness and prevention knowledge, ensuring that students not only recognize how infections spread but also understand the necessary actions to reduce risk.

### 3.5. Barriers to parent-adolescent communication & perception of SRH education

This study examined barriers to parent–adolescent communication regarding sexual and reproductive health (SRH) and student perceptions of SRH education in schools. The findings revealed significant obstacles to open communication, with cultural barriers (45%), parental discomfort (38%), and lack of knowledge (17%) being the most commonly cited reasons for limited discussions Fig 6. A one-way ANOVA test ($F = 21.34, p < 0.001$) confirmed statistically significant differences across various aspects of SRH communication and education perception.

The overall status of communication had a mean score of 1.49 ± 0.50, suggesting that conversations about SRH are rare or inadequate S5 Table. The frequency of communication (1.87 ± 0.33) was low, with many adolescents reporting infrequent or non-existent discussions. The type of communication (1.26 ± 0.44) further indicated that conversations, when they do occur, are often superficial and limited to menstruation and puberty, while topics such as contraception and sexual relationships are frequently avoided. The starting time of SRH communication (1.10 ± 0.30) suggests that discussions typically begin late, often only after menarche or the onset of puberty, missing key opportunities for early education and risk prevention. Additionally, the initiator of SRH communication (1.47 ± 0.50) highlights that parents rarely take the lead, leaving adolescents to seek information from peers, media, or school-based programs.

In contrast, support for SRH education in schools was high, with 72% of students favoring its inclusion, while 28% opposed it Fig 7. Among those in favor, the primary reasons included enhancing knowledge (68%) and preventing risky behaviors (56%), while those opposed cited cultural restrictions (42%) and concerns that it might encourage early sexual activity (31%) (Table 5).

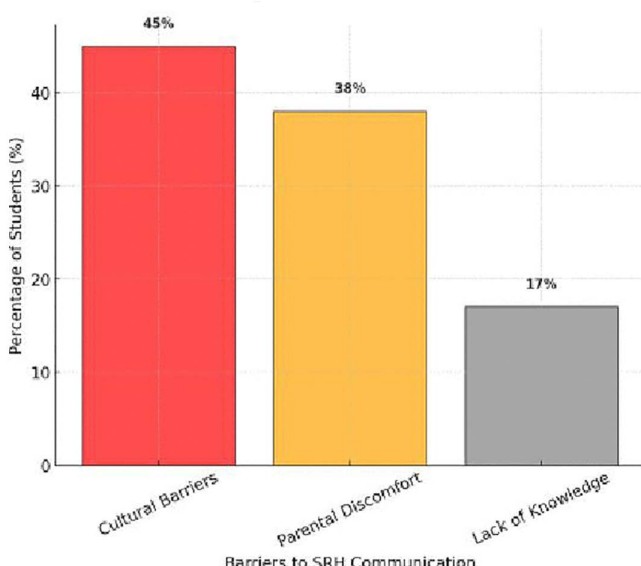

**Fig 6. Barriers to SRH Discussions.** This figure highlights the main barriers to parent–adolescent communication about SRH, including cultural norms, parental discomfort, and lack of knowledge.

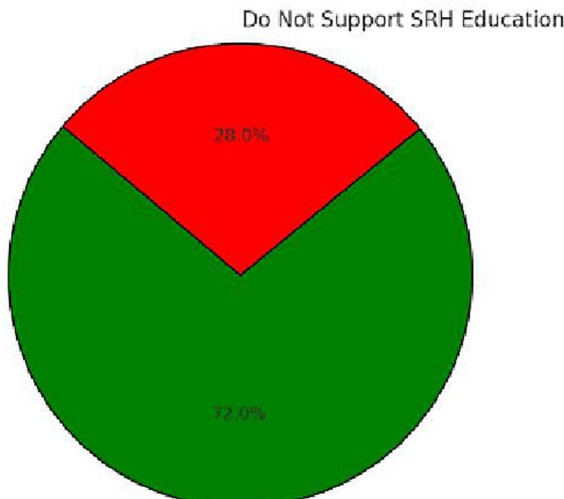

**Fig 7. Support for SRH Education in Schools.** This figure illustrates students' perceptions of school-based SRH education, showing majority support alongside key reasons for both support and opposition.

S6 Fig Highlights that cultural barriers (45%) and parental discomfort (38%) are the primary obstacles to SRH communication, while lack of knowledge (17%) also limits meaningful discussions. S7 Fig shows that 72% of students support SRH education in schools, primarily because they believe it enhances knowledge (68%) and reduces risky behaviors (56%). However, cultural restrictions (42%) and concerns about encouraging early sexual activity (31%) remain major points of opposition.

**Table 5. Comprehensive Barriers to SRH Communication & Education Perception.**

| Category | Mean ± SD | Min Score | Max Score |
|---|---|---|---|
| Overall Status of Communication | 1.49 ± 0.50 | 1.0 | 2.0 |
| Frequency of Communication | 1.87 ± 0.33 | 1.0 | 2.0 |
| Type of Communication | 1.26 ± 0.44 | 1.0 | 2.0 |
| Starting Time of RH Communication | 1.10 ± 0.30 | 1.0 | 2.0 |
| Initiator of RH Communication | 1.47 ± 0.50 | 1.0 | 2.0 |
| Comfort Level in SRH Discussion | 1.78 ± 0.49 | 1.0 | 2.0 |
| Barriers to SRH Communication | Cultural Barriers (45%), Parental Discomfort (38%), Lack of Knowledge (17%) | – | – |
| Support for SRH Education in Schools | Yes (72%), No (28%) | – | – |
| Reasons for Supporting SRH Education | Enhances Knowledge (68%), Prevents Risky Behaviors (56%) | – | – |
| Reasons for Opposing SRH Education | Cultural Restrictions (42%), Encourages Early Sexual Activity (31%) | – | – |

Parent–adolescent communication about SRH remains minimal, with low frequency and late initiation of discussions. Parents often lack the confidence or cultural acceptance to discuss SRH topics, leading adolescents to seek information from unverified sources. Despite cultural resistance, 72% of students supported SRH education in schools, demonstrating the need for structured sexual health curricula. Community-based interventions should be introduced to help parents overcome discomfort and provide accurate SRH information. SRH education programs must be adapted to address cultural concerns while ensuring that students receive medically accurate information.

## 4. Discussion

This cross-sectional survey of female secondary-school students in Al-Madinah Al-Monawara highlights notable gaps in sexual and reproductive health (SRH) literacy. While students demonstrated some understanding of menstruation and reproductive physiology, misconceptions were widespread in the areas of contraception and sexual relationships. These findings echo prior reports from Saudi Arabia, where inconsistent curricula and cultural sensitivities have restricted access to accurate reproductive health education [1,2]. The uneven knowledge profile observed here suggests that adolescents may receive fragmented or partial information, leaving critical domains under-addressed [15,16].

Private school attendance and higher maternal education emerged as independent predictors of higher SRH knowledge scores. These results are consistent with earlier Saudi studies reporting better knowledge among private school students, who may benefit from smaller class sizes, enhanced resources, or more flexible health curricula [2]. The association with maternal education aligns with global evidence showing that educated mothers are more likely to communicate health information and support informed decision-making among their daughters [4,5]. This pattern underscores the intergenerational benefits of female education, suggesting that investment in maternal literacy may indirectly improve adolescent health outcomes [17–21].

Notably, paternal education, age, and family size were not significant predictors in multivariable models. This finding highlights the central role of mothers in reproductive health communication within this context, as also observed in Jordanian and Syrian households, where mothers were more likely than fathers to initiate sensitive conversations [6]. Age-related differences were apparent in univariate analysis, with older students scoring slightly higher, but these differences diminished after adjusting for maternal education and school type, suggesting that structural and familial influences outweigh age alone [22,23].

The study also revealed limited awareness of sexually transmitted infections (STIs), particularly bacterial infections such as gonorrhea and chlamydia. HIV/AIDS was the most frequently recognized, likely reflecting global media coverage and public health campaigns. However, misconceptions were evident, as a substantial proportion of participants believed

casual contact could transmit infections, echoing findings from other conservative settings where stigma and misinformation remain entrenched [9]. Limited awareness of preventive strategies beyond abstinence and monogamy—such as partner screening and routine testing—further underscores the need for comprehensive, evidence-based curricula that address both biomedical and behavioral dimensions of STI prevention [24,25].

Parent–daughter communication emerged as both limited and inconsistent. Fewer than half of participants reported frequent discussions, and cultural barriers, discomfort, and lack of parental knowledge were cited as obstacles. Similar patterns have been documented in Ethiopia and Nepal, where a majority of adolescents report no communication with parents about SRH topics [4,9]. Evidence from a large meta-analysis confirms that open parent–child communication is associated with safer sexual behaviors, including delayed initiation and greater contraceptive use [8]. Strengthening parental capacity to engage in such conversations is therefore a crucial, yet often overlooked, strategy for improving adolescent SRH literacy in conservative societies [26,27].

The findings have several practical implications. First, there is an urgent need to integrate standardized, age-appropriate SRH modules into government school curricula. These modules should address sensitive topics such as contraception and STI prevention with cultural sensitivity, possibly through trained health educators or school nurses. Second, interventions targeting parents, especially mothers, are needed to equip them with accurate information and culturally appropriate communication skills. Parent workshops, online portals, and school-linked education programs could bridge the gap between school learning and home support. Third, awareness campaigns should aim to correct misconceptions, particularly around STI transmission, to reduce stigma and promote healthy behaviors [12,28–31].

### Comparative Overview of Current and Past Studies (2005–2025) on Female Adolescent SRH Literacy

Below is a detailed side-by-side comparison between the current Al-Madinah study and several key peer-reviewed studies from the last 20 years that examined sexual and reproductive health (SRH) knowledge among female adolescents in Saudi Arabia or similar conservative/MENA contexts. Each row outlines the study's reference, setting and sample, population age group, school context, what domains of SRH knowledge were assessed, notable findings (including common misconceptions), parent–child communication patterns, and the study's key conclusions or implications **Table 6**.

### Strengths and limitations

This study benefits from a relatively large sample size, stratified sampling across school types, use of a structured Arabic instrument, and multivariable regression analysis adjusting for key socio-demographic factors. However, several limitations must be acknowledged. The cross-sectional design precludes causal inference. The restriction to female students aged 18 years and above limits generalizability to younger adolescents and male peers. Self-reported responses may be subject to social desirability bias, particularly given the cultural sensitivity of SRH topics. Although the instrument was adapted from validated tools, full psychometric validation within the present sample was limited. Additionally, clustering at the school level was not explicitly modeled, which may slightly underestimate standard errors. Finally, findings from Al-Madinah may not fully generalize to other regions of Saudi Arabia with differing socio-cultural or educational environments.

Assumptions of linear regression were assessed, including normality of residuals and multicollinearity diagnostics.

While students were sampled from multiple schools, clustering at the school level was not explicitly modeled and is acknowledged as a limitation.

## 5. Conclusions

This study demonstrates that adolescent SRH literacy in Al-Madinah is uneven, with private schooling and maternal education serving as important determinants. Misconceptions surrounding contraception, sexual relationships, and STI transmission remain widespread, and parent–daughter communication is constrained by cultural norms and parental

**Table 6. Comparative Overview of Current and Past Studies (2005–2025).**

| Study (Author, Year) | Location & Sample Size | Age Group | School Type | SRH Knowledge Domains Assessed | Key Knowledge & Misconceptions Findings | Parental Communication Patterns | Key Conclusions/ Implications |
|---|---|---|---|---|---|---|---|
| **Current Study (2024)** – Female secondary students in Al-Madinah, Saudi Arabia | Al-Madinah, Saudi Arabia; n = 389 female students (government & private schools) | Secondary (≥18 years old) | Government and private schools | Menstruation, pregnancy, contraception, STIs (HIV/AIDS, etc.) – assessed knowledge of reproductive processes, disease transmission/ prevention | Significant knowledge gaps; misconceptions were prevalent about STIs and contraception (e.g., many wrongly believed casual contact can spread STIs). Private school students had higher knowledge than public school peers, and maternal education correlated with better knowledge. | Mother–daughter communication on SRH was generally limited; many girls relied on peers or the internet for information. However, those with frequent maternal communication had significantly higher knowledge scores. | Urgent need for structured RH education in school curricula and greater parental involvement. Culturally sensitive programs are recommended to dispel myths and improve adolescent girls' SRH literacy. |
| **Alquaiz et al. (2012)** – Riyadh study on sex education knowledge | Riyadh, Saudi Arabia; n = 417 female students (5 classes from 2 schools) | Intermediate & secondary (ages 11–21) | One public and one private school (Riyadh); *no significant knowledge difference by school type* | Sexuality and RH knowledge (puberty, STIs, etc.) and **sources** of information. Assessed recognition of STIs (syphilis, gonorrhea, hepatitis) and general sexual health awareness. | Low STI knowledge, only 33.3% recognized syphilis as an STI, 37.9% gonorrhea, and 14.5% hepatitis. 42% of girls discussed sexual matters with friends vs. just 15.8% with mothers. Teachers were often unapproachable (61% reported teachers had negative attitudes toward sexual health questions) | Very low parent–child communication – only 15.8% discussed sexual topics with a parent (mother). Peers were the main confidants (42%), and even domestic helpers (17.3%) were consulted more than parents for sexual information | Recommended introducing formal sex education in school curricula within cultural/religious context. Emphasized the need for parents and teachers to be more open and proactive in discussing SRH issues with youth. |
| **Tork & Al Hosis (2015)** – RH education intervention study | Qassim, Saudi Arabia; n = 309 female students (3 secondary schools + university preparatory class) | Late adolescence (ages 14–19) | Mixed sample – secondary schools and first-year university (preparatory year) | Puberty and menstrual physiology, pregnancy and antenatal care, and contraception knowledge – assessed before and after a reproductive health education program. | Baseline knowledge deficits were observed (e.g., only ~27% of participants knew about intrauterine device (IUD) contraception pre-intervention); the education program led to significant improvements in knowledge of puberty, menstruation, pregnancy, and contraception (IUD knowledge rose from 27.2% to 67.6% post-intervention). Attitudes toward RH also improved after the training. | Not examined (this study focused on measuring knowledge and attitude changes from the educational intervention, rather than communication patterns). | Demonstrated that a structured RH education program can markedly improve adolescent girls' knowledge in a conservative society. Supports implementing formal RH educational interventions to correct misconceptions and promote informed attitudes. |

*(Continued)*

| Study (Author, Year) | Location & Sample Size | Age Group | School Type | SRH Knowledge Domains Assessed | Key Knowledge & Misconceptions Findings | Parental Communication Patterns | Key Conclusions/ Implications |
|---|---|---|---|---|---|---|---|
| **Gaferi et al. (2018)** – KAP survey in Riyadh schools | Riyadh, Saudi Arabia, $n = 350$ female students from government secondary schools | Secondary school (mid-to-late adolescence, ~15–19 years) | Government public schools (multistage random sample in Riyadh) | General RH knowledge (puberty/menstruation, fertility and ovulation, contraception, STIs) plus menstrual hygiene practices and attitudes | Majority had inadequate knowledge, 66.3% had "inaccurate" overall RH knowledge vs. only 33.7% with correct knowledge.. Specific gaps included poor understanding of fertility (ovulation timing) and low awareness of long-term contraceptive methods. On the positive side, ~95% practiced proper menstrual hygiene and 88% held positive attitudes toward RH topics | Mothers were a primary source of basic puberty information (e.g., 61.2% of girls learned about menstruation from their mother). However, parents rarely discussed sexual topics like STIs – family and even the internet were among the least common sources of STD information. reflecting cultural hesitation to address these issues openly. | Concluded that female adolescents had unsatisfactory knowledge despite generally positive attitudes. Recommended improving adolescents' RH knowledge through school-based programs and by involving parents and teachers in providing appropriate RH education. |
| **Alomair et al. (2022)** – Qualitative study of Saudi women's SRH experiences | Riyadh, Saudi Arabia, $n = 28$ women (married and unmarried, ages 20–50) interviewed in a public hospital. | Adults (20–50) reflecting on their adolescence and beyond (many recalled their teenage SRH knowledge gaps) | N/A (community sample, not school-based) | Explored women's SRH knowledge, perceptions and experiences via themes: menarche experiences, knowledge of sex and reproduction, difficulty discussing SRH, generational gaps, sources of information, and the mother's role | Found a **profound lack of SRH knowledge** among participants, which had led to negative experiences in adolescence and adulthood. Many women had no prior understanding of menstruation (causing fear/distress at menarche) and held deep-rooted negative views about sex, contributing to physical and psychological issues in marriage. Misinformation was common and formal education on these topics was absent. | Almost all women reported receiving little to no information from parents or teachers during their youth. SRH topics were socially taboo ("a culture of shame"), so as adolescents they resorted to covert sources – e.g., peers or later, the internet – to learn, often encountering myths in the process. | Highlighted an urgent unmet need for culturally tailored SRH education for Saudi youth and women. Recommended developing formal SRH programs that address socio-cultural barriers and leveraging reliable media (e.g., Arabic web resources) to provide evidence-based information, thereby improving women's SRH literacy. |

*(Continued)*

| Study (Author, Year) | Location & Sample Size | Age Group | School Type | SRH Knowledge Domains Assessed | Key Knowledge & Misconceptions Findings | Parental Communication Patterns | Key Conclusions/ Implications |
|---|---|---|---|---|---|---|---|
| **Jaffer et al. (2005)** – National adolescent KAP survey in Omanapplications.emro. who.int | Oman (national sample), $n \approx 1,675$ female secondary-school students (plus ~1,670 male students) applications. emro.who.int | Secondary school (mostly ages ~15–18) | Nationally representative government schools (multiple regions) | Puberty knowledge (physical changes in self vs. opposite sex), attitudes on marriage timing and desired family size, knowledge/attitudes on contraception ("birth spacing"), knowledge of fertile period, and awareness of HIV/AIDS and other STIs,also attitudes toward female genital cutting (FGC) applications.emro. who.intapplications. emro.who.int. | **Substantial knowledge gaps**, only ~50% of girls knew the pubertal changes of their **own** sex (and even fewer understood boys' puberty)applications.emro. who.int. Knowledge of the fertile period in the menstrual cycle was generally poor, as was knowledge of HIV/AIDS and STI transmissionapplications.emro. who.int. Nonetheless, about two-thirds expressed positive attitudes toward modern contraceptives and intended to use them in the future. Alarmingly, 80% of adolescents (boys *and* girls) approved of FGC – reflecting a prevalent misconception or traditional norm in the societyapplications.emro. who.int. | Parent–child communication was not explicitly measured in this survey. The widespread lack of basic puberty and sexual health knowledge suggests that many youth did **not** receive adequate information at home or school before encountering these issues. (Notably, the study asked if adolescents had received information *before* puberty, many had not, contributing to confusion and myths.) | Underscored critical gaps in adolescents' RH literacy in a conservative society. These findings imply a need for integrating comprehensive RH education for youth in Oman. Addressing misbeliefs (e.g., high support for FGC and poor STI knowledge) is important for public health, the authors urged policymakers to consider youth-focused RH programs to improve knowledge and correct harmful misconceptions. |
| **Othman et al. (2020)** – Parent focus groups in Jordan | Amman & northern Jordan, $n = 20$ focus group discussions with parents (Jordanian and Syrian participants) | Parents (mostly in 30s–40s) of adolescent children | N/A (community setting across conservative communities) | Explored parent–child SRH communication: which topics are discussed, cultural taboos, and strategies parents use to convey (or avoid) SRH information to sons and daughters. | Parents acknowledged a prevailing *"culture of shame"* that makes direct SRH discussions difficult. Nonetheless, many expressed willingness to break these taboos and identified practical strategies: [1] using gender-matched communication (fathers with sons, mothers with daughters), [2] mothers serving as a comfortable "safe space" for sensitive discussions, and [3] seeking help from others – either involving trusted relatives or relying on school-based SRH education to fill the gap. Each approach had perceived strengths and challenges, and topics discussed varied by strategy. | Mother–daughter communication was seen as especially critical (mothers often took responsibility for educating daughters, while fathers talked to sons) However, many parents still avoided detailed sexual topics due to embarrassment, sometimes deferring to schools or healthcare providers. Overall, open dialogue was limited and usually segregated by parent/ child gender. | Emphasized the need to support and educate parents for better SRH communication with adolescents. Interventions were suggested to build parents' knowledge and confidence in discussing sexual topics. The study recommended leveraging culturally acceptable methods – e.g., gender-matched conversations, school-based programs, and guidance from religious/community institutions – to help **de-stigmatize** SRH |

discomfort. Interventions that combine school-based curricula with family-oriented communication supports are needed to ensure adolescents acquire the knowledge and confidence required for healthy sexual and reproductive lives.

## Author contributions

**Conceptualization:** Asma Alshanqiti.

**Data curation:** Asma Alshanqiti.

**Formal analysis:** Asma Alshanqiti, Taif N. Alahmadi.

**Investigation:** Elaf Aljabri, Aisha R. Al-Rashidi, Taif N. Alahmadi, Raghdah Alrehaili.

**Methodology:** Asma Alshanqiti, Elaf Aljabri, Aisha R. Al-Rashidi, Taif N. Alahmadi, Raghdah Alrehaili.

**Project administration:** Raghdah Alrehaili.

**Resources:** Reem F. Al-Mughthawi, Fatema A. Saleh, Raghdah Alrehaili.

**Software:** Reem F. Al-Mughthawi, Fatema A. Saleh.

**Supervision:** Asma Alshanqiti, mohamed wagdy.

**Validation:** Fatema A. Saleh.

**Visualization:** Elaf Aljabri, Fatema A. Saleh.

**Writing – original draft:** Elaf Aljabri, Aisha R. Al-Rashidi.

**Writing – review & editing:** Asma Alshanqiti, Aisha R. Al-Rashidi, mohamed wagdy.

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
