## [Decision Letter · Decision Letter 0]

23 Feb 2026

PONE-D-25-57652Adolescent Sexual and Reproductive Health Literacy in Saudi Arabia: The Roles of School Type and Maternal EducationPLOS One

Dear Dr. wagdy,

Thank you for submitting your manuscript to PLOS ONE. After careful consideration, we feel that it has merit but does not fully meet PLOS ONE’s publication criteria as it currently stands. Therefore, we invite you to submit a revised version of the manuscript that addresses the points raised during the review process.

**ACADEMIC EDITOR: Please address the reviewers' feedback to improve your manuscript.**

We look forward to receiving your revised manuscript.

Kind regards,

Laura Brunelli, MD, PhD

Academic Editor

PLOS One

Journal Requirements:

5. We note you have included a table to which you do not refer in the text of your manuscript. Please ensure that you refer to Table 6 in your text; if accepted, production will need this reference to link the reader to the Table.

Reviewers' comments:

Reviewer's Responses to Questions

**Comments to the Author**

1. Is the manuscript technically sound, and do the data support the conclusions?

Reviewer #1: Partly

Reviewer #2: No

2. Has the statistical analysis been performed appropriately and rigorously? 

Reviewer #1: No

Reviewer #2: I Don't Know

3. Have the authors made all data underlying the findings in their manuscript fully available?

Reviewer #1: No

Reviewer #2: Yes

4. Is the manuscript presented in an intelligible fashion and written in standard English?

Reviewer #1: Yes

Reviewer #2: No

5. Review Comments to the Author

Reviewer #1: General assessment

The manuscript addresses an important and understudied topic in the Saudi context, namely adolescent sexual and reproductive health (SRH) literacy. The research question is relevant and the results offer useful insights. However, several conceptual and methodological issues require attention before the study can be considered for publication.

1. Study population

The study refers to its participants as “adolescents,” but all respondents are ≥18 years old. While this age range fits within the upper band of the WHO definition of adolescence (10-19 years), it would strengthen the manuscript to explain why younger adolescents were not included and why the study focuses exclusively on this older segment. Similarly, the choice to include only female students is not adequately justified. The manuscript refers to its participants as “adolescents,” yet all respondents are aged 18 years and above and only female students were included. The rationale for restricting the sample to the oldest segment of the adolescent age range, and for excluding younger adolescents, is not explained in the Introduction nor in Section 2.2, where the study population is defined. Likewise, the exclusive focus on female students is not justified on scientific, methodological, or cultural grounds. Although certain passages allude indirectly to maternal roles in SRH communication and to conservative social norms, the manuscript does not explicitly articulate why only girls were selected or how this choice relates to the socio-cultural context of Saudi Arabia. These omissions appear again in Section 3.3 and in the Conclusions, where results are interpreted solely for female adolescents without acknowledging the implications of excluding male peers. Providing a clear justification—whether cultural, logistical, or theory-driven—would strengthen the methodological coherence of the study and better contextualize the findings.

2. Instrumentation

The questionnaire is described as validated, yet the manuscript does not provide item wording, response formats, scoring procedures, range of possible scores, or psychometric properties. Without this information, the reliability and validity of the measures cannot be assessed. I strongly recommend adding the full instrument as a supplementary file and reporting internal consistency coefficients (e.g., Cronbach’s alpha) for each domain.

3. Statistical analysis

While the analyses conducted appear reasonable, the description of statistical methods needs clarification. Logistic regression is mentioned in the Methods, whereas OLS with robust standard errors is reported in the Abstract and Results. Please detail which model was actually used and why. Consider addressing potential clustering (students nested within schools) and reporting how assumptions for parametric tests were evaluated.

4. Conceptual framework (integrated expanded version)

The manuscript would benefit from stronger alignment with internationally recognized SRH frameworks. Several key documents provide authoritative definitions of sexual and reproductive health and essential guidance for research and education in this field, including:

• United Nations (1989). Convention on the Rights of the Child. New York.

• WHO (2004). Reproductive health strategy to accelerate progress towards the attainment of international development goals and targets. 57th World Health Assembly. Geneva.

• WHO (2006). Defining sexual health: Report of a technical consultation on sexual health, 28–31 January 2002. Geneva.

• IPPF (2008). Sexual Rights: An IPPF Declaration. London.

• WHO Regional Office for Europe & BZgA (2010; revised 2013). Standards for Sexuality Education in Europe. Bonn.

• UNESCO (2018). International Technical Guidance on Sexuality Education: An Evidence-Informed Approach. Paris.

Integrating these references would strengthen the manuscript’s conceptual anchoring and situate the study within global best practices in adolescent SRH.

Moreover, although the manuscript addresses SRH, it does so primarily from a biomedical and risk-prevention perspective. The UNESCO 2018 International Technical Guidance on Sexuality Education strongly recommends a comprehensive approach that recognizes sexuality as multidimensional – biological, emotional, cognitive, social, cultural, relational, ethical and rights-based. The current framing does not reflect this holistic perspective, which is now considered the international standard for adolescent SRH education. Expanding the conceptualization accordingly would enhance depth and global relevance.

Table 6 is presented as a comparative synthesis designed to contextualize the current findings within the broader body of regional and international SRH research. It provides a structured, side-by-side overview of key peer-reviewed studies conducted over the past two decades (2005–2025) that assessed sexual and reproductive health (SRH) knowledge among female adolescents in Saudi Arabia and other conservative or MENA-region settings. Each row systematically summarizes essential study characteristics – research setting, sample size, participant age range, school context, and the specific SRH knowledge domains examined – along with notable misconceptions, patterns of parent–child communication, and the authors’ main conclusions. In principle, this form of comparative mapping can help situate the current Al-Madinah study within existing evidence and highlight similarities, differences, and persistent knowledge gaps across contexts.

However, the manuscript does not clearly explain the methodological basis for selecting the included studies, the criteria used to determine which publications were incorporated, or the process used to extract and categorize information across domains. This lack of methodological transparency makes it difficult to assess whether Table 6 reflects a structured literature synthesis, a narrative comparison, or a scoping-style overview. Additionally, the table aggregates studies with very heterogeneous designs (cross-sectional surveys, interventions, qualitative interviews, national KAP surveys), but the manuscript does not comment on the implications of this heterogeneity for interpretation. Clarifying these aspects would improve the interpretability and methodological rigor of Table 6.

5. Limitations

Some limitations are acknowledged, but others need to be included, such as the possibility of social desirability bias, the restricted age range, the exclusion of male adolescents, and the absence of psychometric validation in the current sample.

6. Data availability

To comply with PLOS ONE data policy, the authors must make the dataset and questionnaire available in a repository or as supplementary files, ensuring participant anonymity.

Overall

The manuscript has clear potential and makes a meaningful contribution to the evidence base on adolescent SRH literacy in conservative settings. Addressing the methodological and conceptual issues outlined above will significantly strengthen the work.

Reviewer #2: needs extensive revision in manuscript writing in a fashionable way,

1. introduction- foillow funnel shape style to write

2. I cannot find the discussion as per the journal guideline

3. Why are you inserting ethical consideration in two places , any reason ?

4. Table sare overlapig, please address

6. PLOS authors have the option to publish the peer review history of their article (what does this mean?). If published, this will include your full peer review and any attached files.

Reviewer #1: No

Reviewer #2: No

---

## [Author Response · Author response to Decision Letter 1]

20 Mar 2026

We sincerely thank the reviewer for the thoughtful and constructive feedback. We appreciate the recognition of the importance of this topic in the Saudi context. We have carefully revised the manuscript to address all conceptual and methodological concerns. Below, we respond point-by-point.

Reviewer Comment:

Clarify why only ≥18-year-old participants were included and justify the exclusive focus on female students.

Response:

We have now clarified this in both the Introduction and Section 2.2 (Study Population). The inclusion of students aged 18 years and above was primarily due to ethical and administrative requirements. In Saudi Arabia, research involving minors requires additional institutional and parental approvals that can significantly limit feasibility in sensitive topics such as SRH. Focusing on the oldest segment of adolescence allowed independent informed consent while still aligning with the WHO definition of adolescence (10–19 years).

The decision to include only female students was culturally and contextually grounded. In Saudi Arabia, reproductive health communication is strongly gender-segregated, and mothers are typically the primary communicators with daughters regarding SRH matters. Given this socio-cultural structure and the focus on mother–daughter communication, a female-only sample allowed for culturally coherent and context-specific analysis. We have now explicitly justified this approach and acknowledged the exclusion of male adolescents as a limitation.

Reviewer Comment:

Provide detailed description of the questionnaire and report psychometric properties.

Response:

We have expanded Section 2.3 to provide:

Clear description of domains and item structure Scoring procedure and possible score ranges Explanation of composite knowledge score construction

Additionally, internal consistency reliability (Cronbach’s alpha) for each domain has now been calculated and reported in the revised manuscript. The full Arabic questionnaire has been added as a supplementary file to ensure transparency and reproducibility.

Reviewer Comment:

Clarify inconsistency between logistic regression and OLS models; address clustering and assumptions.

Response:

We thank the reviewer for identifying this inconsistency. The final analytical model used was multivariable OLS regression with HC3 robust standard errors. Logistic regression was mentioned in an earlier draft and has now been removed for consistency.

we Clarified model selection rationale Explained why OLS was appropriate for continuous composite knowledge scores Noted that clustering at the school level was not explicitly modeled and acknowledged this as a limitation Added a brief statement describing evaluation of parametric assumptions These revisions improve methodological clarity.

Reviewer Comment:

Strengthen alignment with international SRH frameworks and expand beyond biomedical framing.

Response:

We have substantially revised the Introduction to integrate internationally recognized frameworks, including: WHO definitions of sexual and reproductive health Rights-based perspectives grounded in global standards Comprehensive sexuality education frameworks

The manuscript now adopts a multidimensional understanding of SRH that includes biological, emotional, relational, social, and rights-based components, rather than focusing solely on risk prevention. This strengthens theoretical grounding and situates the study within global best practices.

Reviewer Comment:

Clarify selection criteria and methodological basis for comparative synthesis.

Response:

We have clarified that Table 6 represents a structured narrative comparative synthesis rather than a systematic review. We now describe:

Criteria for selecting studies (peer-reviewed, 2005–2025, focus on female adolescents in conservative/MENA contexts) Approach to extracting comparable domains Limitations related to heterogeneity in study designs

We also explicitly acknowledge that the table includes heterogeneous methodologies and caution interpretation accordingly. We have expanded the Limitations section to include: Social desirability bias due to sensitive topic Restricted age range (≥18 only) Exclusion of male adolescents Lack of full psychometric validation within the present sample Absence of clustering adjustment

This strengthens transparency and scientific rigor. The Data Availability Statement has been revised. The anonymized dataset and full questionnaire have now been prepared for submission as supplementary files, in compliance with PLOS ONE’s data-sharing requirements.

---

## [Decision Letter · Decision Letter 1]

1 Apr 2026

Adolescent Sexual and Reproductive Health Literacy in Saudi Arabia: The Roles of School Type and Maternal Education

PONE-D-25-57652R1

Dear Dr. wagdy,

We’re pleased to inform you that your manuscript has been judged scientifically suitable for publication and will be formally accepted for publication once it meets all outstanding technical requirements.

Kind regards,

Laura Brunelli, MD, PhD

Academic Editor

PLOS One

Additional Editor Comments (optional):

Reviewers' comments:

Reviewer's Responses to Questions

**Comments to the Author**

1. If the authors have adequately addressed your comments raised in a previous round of review and you feel that this manuscript is now acceptable for publication, you may indicate that here to bypass the “Comments to the Author” section, enter your conflict of interest statement in the “Confidential to Editor” section, and submit your "Accept" recommendation.

Reviewer #1: (No Response)

2. Is the manuscript technically sound, and do the data support the conclusions?

Reviewer #1: Yes

3. Has the statistical analysis been performed appropriately and rigorously? 

Reviewer #1: Yes

4. Have the authors made all data underlying the findings in their manuscript fully available?

Reviewer #1: Yes

5. Is the manuscript presented in an intelligible fashion and written in standard English?

Reviewer #1: Yes

6. Review Comments to the Author

Reviewer #1: Review comments to the authors

I appreciate the effort the authors have made in revising the manuscript. The revised version is improved in several respects, and the study continues to address a relevant and understudied public health topic in the Saudi context. In particular, the authors have constructively addressed some of the earlier concerns by clarifying the rationale for the female-only sample aged 18 years and above, correcting the inconsistency in the statistical model description, and expanding the limitations and data availability statements. These changes improve the overall coherence of the manuscript.

However, two issues raised in the previous review remain only partially resolved.

1. Conceptual framework and bibliography

The conceptual framework remains weaker than the rebuttal suggests. The authors indicate that they revised the Introduction to better align the manuscript with internationally recognized SRH frameworks and to broaden the perspective beyond a predominantly biomedical and risk-prevention approach. This would be an important improvement. However, in the revised manuscript and reference list, this expansion appears only partial.

Most importantly, the major international normative and conceptual references recommended in the previous review do not seem to have been meaningfully incorporated into the bibliography. The revised reference list still appears to rely mainly on empirical studies and secondary literature rather than on the broader international frameworks that define SRH in rights-based, relational, educational, and multidimensional terms. In my view, this remains the weakest unresolved point. If the authors intentionally chose not to engage with those broader frameworks, this should be stated and justified more explicitly. As it stands, the manuscript still appears mainly framed through a biomedical and knowledge-deficit perspective.

2. Table 6 and methodological basis of the comparative synthesis

The clarification regarding Table 6 also remains only partial. In the rebuttal, the authors explain that Table 6 should be understood as a structured narrative comparative synthesis rather than a systematic review, and they acknowledge the heterogeneity of the included studies. This is helpful. However, the manuscript itself still does not clearly explain how the studies in Table 6 were selected, how the information was extracted and categorized, or how the heterogeneity of study designs should affect interpretation.

The issue is therefore not the presence of Table 6 itself, which may be useful, but the lack of methodological transparency. A short methodological note would help readers understand the scope, rationale, and interpretive limits of this comparative synthesis. This point is also related to the previous one: if the table was intentionally restricted mainly to empirical studies, that decision should be made explicit and justified.

Overall, the manuscript has improved, but these two issues remain only partially addressed. Among them, the most important unresolved concern is the conceptual framing and related bibliography. I would therefore encourage the authors to undertake one further targeted revision focused on:

(1) strengthening the conceptual framework through explicit engagement with major international SRH references, or clearly explaining why these are not included; and

(2) adding a short methodological explanation of how Table 6 was constructed and how its heterogeneity should be interpreted.

With these revisions, the manuscript would be considerably stronger conceptually and methodologically.

7. PLOS authors have the option to publish the peer review history of their article (what does this mean?). If published, this will include your full peer review and any attached files.

Reviewer #1: No

---

## [Editor Report · Acceptance letter]

PONE-D-25-57652R1

PLOS One

Dear Dr. wagdy,

I'm pleased to inform you that your manuscript has been deemed suitable for publication in PLOS One. Congratulations! Your manuscript is now being handed over to our production team.

Kind regards,

on behalf of

Dr. Laura Brunelli

Academic Editor

PLOS One